# Fungi, P-Solubilization, and Plant Nutrition

**DOI:** 10.3390/microorganisms10091716

**Published:** 2022-08-26

**Authors:** Maria Vassileva, Gilberto de Oliveira Mendes, Marco Agostino Deriu, Giacomo di Benedetto, Elena Flor-Peregrin, Stefano Mocali, Vanessa Martos, Nikolay Vassilev

**Affiliations:** 1Department of Chemical Engineering, University of Granada, C/Fuentenueva s/n, 18071 Granada, Spain; 2Laboratório de Microbiologia e Fitopatologia, Instituto de Ciências Agrárias, Universidade Federal de Uberlândia, Monte Carmelo 38500-000, Brazil; 3PolitoBIOMed Lab, Department of Mechanical and Aerospace Engineering, Politecnico di Torino, 10129 Turin, Italy; 4Enginlife, 10128 Torino, Italy; 5Council for Agricultural Research and Analysis of the Agricultural Economy, Research Centre for Agriculture and Environment, 50125 Firenze, Italy; 6Institute of Biotechnology, University of Granada, 18071 Granada, Spain

**Keywords:** sustainable agriculture, fungi, P-solubilization, alternative P-sources, new strategies for P-solubilization

## Abstract

The application of plant beneficial microorganisms is widely accepted as an efficient alternative to chemical fertilizers and pesticides. It was shown that annually, mycorrhizal fungi and nitrogen-fixing bacteria are responsible for 5 to 80% of all nitrogen, and up to 75% of P plant acquisition. However, while bacteria are the most studied soil microorganisms and most frequently reported in the scientific literature, the role of fungi is relatively understudied, although they are the primary organic matter decomposers and govern soil carbon and other elements, including P-cycling. Many fungi can solubilize insoluble phosphates or facilitate P-acquisition by plants and, therefore, form an important part of the commercial microbial products, with *Aspergillus*, *Penicillium* and *Trichoderma* being the most efficient. In this paper, the role of fungi in P-solubilization and plant nutrition will be presented with a special emphasis on their production and application. Although this topic has been repeatedly reviewed, some recent views questioned the efficacy of the microbial P-solubilizers in soil. Here, we will try to summarize the proven facts but also discuss further lines of research that may clarify our doubts in this field or open new perspectives on using the microbial and particularly fungal P-solubilizing potential in accordance with the principles of the sustainability and circular economy.

## 1. Introduction

The continuously growing human population determines the increased global demand for high agricultural productivity, which, at the same time, should follow the principles of sustainability and circular economy. In the last 15 years, several biotechnological approaches were developed to enhance plant growth and health with safe and environmentally mild alternatives, including those based on microorganisms. The practical realization of the proposed strategies will significantly reduce the indiscriminate use of chemical fertilizers and pesticides. Despite the environmentally harsh conditions of surviving, soil contains many individual microbial taxa, including different members of the three domains of life. Although we still sub-estimate (for methodological reasons) microbial diversity [1], microorganisms are one of the key components of both natural and cultivated soils thus affecting the soil quality and plant productivity [2]. It should be noted that long-term chemical fertilization and application of pesticides decreased both the soil microbial species richness and microbial–plant beneficial interactions as a part of the plant holobiont [3]. Therefore, rebuilding soil productivity by applying bioeffectors (biostimulants or biofertilizers) is a priority and one of the most studied biotechnological alternatives to chemical fertilizers.

The term biofertilizer has different definitions but in general, it includes plant beneficial microorganisms and their derivates (metabolites), excluding biocontrol agents [4,5,6,7]. Among beneficial microorganisms, bacteria and fungi are considered the most important in helping plant nutrient acquisition and improving plant health. In general, the co-existence of fungi and bacteria is reported in microbiomes with different profile characteristics (animal, soil, and food microbiomes) [8]. Particularly in soils, they are reported as the most abundant microorganisms with 10^2^–10^4^ times more biomass than protists, archaea and viruses [1]. Amongst various functions that bacteria and fungi perform in soil ecosystems, particularly important is their contribution to plant growth and development, and plant diversity. It was demonstrated that fungi play an important role in the utilization of easily available and more complex litter-derived C than bacteria, thus actively taking part in soil formation [9]. However, both bacteria and fungi are present in the soil microbial hotspots where the soil organic carbon decomposition is much higher than in the bulk soil [10].

It was shown that annually, mycorrhizal fungi and nitrogen-fixing bacteria are responsible for 5 to 80% of all nitrogen, and up to 75% of P plant acquisition [2]. However, while bacteria are the most studied soil microorganisms and most frequently reported in the scientific literature, the role of fungi is relatively understudied [11], although they are the primary organic matter decomposers and govern soil carbon and other elements cycling [12]. It is also well established that fungi, particularly in the zone with an abundant presence of fungal hyphae or roots and hyphae (mycosphere or mycorrhizosphere, respectively), greatly affect bacterial growth in soil and, consequently, their interactions with plants [13]. The potential rapid growth and distribution of a given functional bacterium within the community are often linked to fungi as mediators of ecological processes which also impact the diversity of bacterial communities [14]. On the other hand, the cooperation between fungi and bacteria is a selective process depending on the soil, although this phenomenon should be further studied in soil and other systems as well [13,14].

Fungi are known as a diverse and multifunctional group of soil microorganisms, which demonstrate a high capacity to adapt to various adverse abiotic conditions such as salinity, drought, heavy metals, and extreme pH [15,16,17]. It is also important to mention that fungi manifest significant tolerance to low water activity (a_w_) values and high osmotic pressure as proved when growing on solid substrates [18], preserving at the same time high metabolic activity. In soil, 1.5 million fungal species can be found free-living in the bulk soil or as endophytes occupying plant tissues, the mycorrhizae being the most studied beneficial fungus–plant association [19].

Many fungi are able to solubilize insoluble phosphates or facilitate P-acquisition by plants and, therefore, form an important part of the commercial microbial products, with *Aspergillus*, *Penicillium* and *Trichoderma* being the most efficient. In this paper, the role of fungi in P-solubilization and plant nutrition will be presented with a special emphasis on their production and application. Although this topic has been repeatedly reviewed, some recent views questioned the efficacy of the microbial P-solubilizers in soil. This short review opinion will try to summarize the proven facts, but also discuss further lines of research that may clarify our doubts in this field or open new perspectives on using the microbial and particularly fungal P-solubilizing potential in accordance with the principles of sustainability and circular economy.

## 2. P-Bearing Sources

### 2.1. Conventional P-Sources

The amount of P that is available to plants in cultivable soils is frequently low [20]. To satisfy the need of the plants, the P is added to soil in the form of phosphate fertilizers, but the overall P use efficiency is low because, although plants utilize a fraction of soluble P, the rest rapidly forms insoluble complexes with soil constituents [21]. Therefore, frequent application of soluble forms of inorganic P is normally above what would be necessary under ideal conditions. Even under adequate P-fertilization, only a fraction of the applied P is acquired by the first year’s plant growth [22]. A part of the chemical P-fertilizer can be converted into sparingly soluble calcium (alkaline soils), aluminum, and iron (acidic soils) salts of P or be fixed to soil minerals. It was estimated that in the middle of this century about 14 million tons of phosphate fertilizers will be applied, seven million of which will remain in the soil [22].

The basic raw material to produce phosphate fertilizers is rock phosphate composed mainly of the phosphate mineral apatite [Ca_5_(PO_4_)_3_(Cl/F/OH)]. The rock phosphate is processed to remove the bulk of the contained impurities, resulting in a concentrate with a content of P_2_O_5_ ranging from 26% to 34% and up to as much as 42%. It should be noted that lower P-concentrations in rock phosphate and lower quality deposits generate more waste materials, and, on the other hand, more energy and chemicals are required per ton of useful phosphate produced [23]. It is also important to mention the great risk of contamination with different metals present in varying concentrations in rock phosphate and its fertilizer derivates [24,25]. Three facts should be considered when assessing the current situation in this field. One of the main reasons for concern in both phosphate rock-mining and P-fertilizer industries is that inexpensive, high-grade rock phosphate reserves could be exhausted in the next 60 to 80 years as phosphate-bearing ore is a finite non-renewable resource [21,23,26]. P-fertilizer use has increased by around 2.5% per year, 4- to 5-fold during the last 50 years and is projected to increase in the first three decades of the 21st century by 20 Tg per year [27]. Second, there are no substitutes for phosphorus in agriculture, but, on the other hand, phosphorus is recycled by using animal manure and sewage sludge (although there are serious concerns about these alternatives). Finally, it is important to note that the cultivation soil available per capita on a global level is lowering while the world population is increasing by 250,000 people every day (approximately 80 million per year). This situation will result in the enhanced need for food and, consequently, phosphate demand will increase at a rate of 1.5–2% [27]. Therefore, there is an urgent need to find novel phosphate sources and/or novel P-solubilization techniques for ensuring phosphate availability.

### 2.2. Alternative P-Sources

#### 2.2.1. Struvite

Here, the emphasis will be on potential P-bearing sources such as struvite and bone char. Struvite is a P recovery product, and it should be mentioned that scientific attention to the recovery of P is increasing because of its agricultural and industrial importance and bearing in mind the rapid depletion of natural P-resources. Struvite (NH_4_MgPO_4_·6H_2_O) is relatively abundant in soils and lakes and can be found naturally formed through a variety of reactions with sources such as bird droppings and fish bones, but also in water treatment plants in form of crystals in wastewater pipes [28,29]. Controlled struvite formation can be carried out in reactors during the sludge digestion process thus forming a magnesium ammonium phosphate fertilizer for its use in agriculture [30]. It is also important to note that the NH_4_^+^ of the struvite composition can be readily replaced by potassium, thus forming MgKPO_4_·6H_2_O (potassium struvite) [31]. This form of struvite could serve as a source of different important nutrients, which enhance its fertilizing value. As with all other P-bearing natural sources, struvite can be added to soils without any previous treatment, but its fertilizing effect is not well pronounced. When assessing the agronomic value of struvite, it should be also mentioned that the solubility of different sources of struvite depends on their chemical and physical composition and soil characteristics such as pH [32]. Application of soluble P during early plant growth is crucial to determine high growth and plant development with optimal yield. However, struvite added to soil–plant systems is not able to satisfy the need of P during this phase of the plant growth as it is only slightly soluble in water (1–5 %) [33]. Even when applied to soil in combination with soluble chemical P-fertilizers, dissolution of struvite is not registered and therefore its presence is not beneficial to early plant growth [34]. In general, struvite solubility is very low in water, gradually decreasing from pH 7.0 to 8.5 in calcareous soils with a high pH [35]. Therefore, it is important to study how to increase the struvite solubility to meet the plant P demand, particularly in the early stage of plant growth during the establishment of the root system, always bearing in mind that this slow-release fertilizer has a P-concentration similar to different superphosphates [36]. Application of *Aspergillus niger* is an option to increase the solubility of struvite, releasing soluble P [37]. A general scheme for struvite solubility and functionality depending on the type of soil, soil conditions (pH, presence of organic acids), characteristics of the struvite related to its nature and formation mode could serve when assessing the nutritional need of the respective crop to optimize its yield.

#### 2.2.2. Biochar and Bone Char

Two main types of biochar could be distinguished, depending on the origin of the material before the treatment: biochar and bone char. Biochar is a biologically derived material produced after a thermal degradation of organic materials such as agricultural wastes. The production of biochar in the absence or very little presence of oxygen is now a very used process mainly due to the energy potential of the respective organic sources [38]. Particularly interesting and well-studied is the pyrolysis of agricultural crop residues producing biochar, which is further applied as a soil amendment [39]. In soil, biochar amendment is accepted as a sustainable approach with multifaceted benefits starting with the management of agricultural wastes, bioenergy production, carbon sequestration during biochar production, improving soil biological and physical characteristics, improving resistance to diseases, and promoting the growth of plants [40]. It was reported that biochar application positively affects the stress tolerance of plants to different abiotic factors derived from industrial activities or climate change such as salinity, drought, metal toxicity and high temperature [41,42]. Another important benefit of applying biochar is its P-content. It was suggested that recycling agricultural residues through biochar may improve sustainable P-recycling bearing in mind the enormous number of agricultural wastes and particularly manures, which have four times higher P-content compared to some solid wastes [43]. Independently of the form of P in the wastes, biochar is much richer in P, which is always insoluble. During the thermal treatment of biochar, the P-content ends in insoluble complexes with Al, Fe, Ca, Mg, and other metal cations [44].

Bone char is another potential alternative source of P and particularly in low-income countries is considered a very attractive slow-release P-bearing source [45]. Like in the case of biochar, bone char is produced via the pyrolysis process, in which bone-containing material is treated in absence of oxygen and at temperatures ranging from 200 to 700 °C. Until recently, residual materials of meat production were efficiently used as high-protein components of animal feed. However, because of the bovine spongiform encephalopathy crisis in the European meat industry, this use of animal wastes is now strictly controlled. The main treatment options are incineration and pyrolysis, which sterilize the product, preventing the transmission of diseases associated with raw animal products [46], and increase the concentration of desirable nutrients such as P and calcium (up to 47%) producing bioapatite and huge amounts of ash, the valorization of which is a major concern. Therefore, apart from using bone residues for their energy content (~17,000 kJ kg^−1^), these wastes can be applied to soil as part of the sustainability strategies of the fertilizer industry and agriculture [47]. An important advantage of this P-bearing source is its purity compared to rock phosphate, as it is almost totally free of heavy metals and radionuclides. For example, the cadmium concentrations in bone char range up to only 3.03 mg kg^−1^ in cattle and pig bone but up to 556 mg kg^−1^ in rock phosphate [48].

Both bio and bone char are sources of an inorganic, insoluble form of P. Here, P-solubilizing microorganisms could be tested and used as a tool for high and rapid solubilization in fermentation systems or in soil [49,50]. To facilitate the microbial P-solubilizing function and due to their highly porous structure, bone char and biochar were used as soil amendments and simultaneously as cell/spore carriers [49,51]. Biochar derived from agricultural wastes, including bones, can be considered a potential carrier for the formulation of microbial inoculants and might replace other cheap and widely used commercial materials such as peat. It should be noted that, when assessing the role of biochars as potential P-sources and soil P-improvers, we should also distinguish between different types of these products formed at different temperatures, 500–700 °C being the most appropriate [49,52].

## 3. Microbial Solubilization of P-Bearing Sources

If some of the conventional and alternative P-sources are applied directly to soil without previous treatment, even with soil acidity below pH 5.5–6.0, they will become as effective as superphosphate after only 4 years of annual direct application [53,54]. It is well established that the application of apatite-based P-sources (rock phosphate and biochars) is not economically feasible, particularly in soil conditions characterized by a high P-fixation capacity [55]. In recent years, various techniques for phosphate solubilization have been proposed, with increasing emphasis on the application of P-solubilizing microorganisms. Microbial P-solubilization is one of the most studied characteristics of soil microorganisms, particularly because of the interest in the production of biofertilizers [7]. Since the early work of Sperber [56], it is widely accepted that the microbial P-solubilization process is based on the organic acid production or release of protons, which attack the structure of the inorganic P-sources. Chelation is a particularly important tool for inorganic P-solubilization by microbially produced organic acids that, through their hydroxyl and carboxyl groups, chelate the cations (mainly calcium) bound to phosphate, the latter being converted into soluble forms [57]. The number of carboxylic groups is of great importance and determines the efficacy of solubilization. Tri- and di-carboxylic acids, such as citric and oxalic, were shown to be very strong solubilizers compared to monocarboxylic acid (gluconic acid) [58,59]. Moreover, it is known that Ca^2+^ forms bidentate; mononuclear complexes with two carboxylic acid groups [60]. Fungal-formed Ca-oxalates are found everywhere in the environment. The production of oxalic acid is a natural process, which can increase the rate of soil weathering, enhancing the availability of nutrients including P [61]. Many free-living and symbiotic fungi can solubilize inorganic phosphate and increase phosphorus availability for plants [62,63] including in highly weathered soils [64] (Figure 1).

Two types of fungal P-solubilization could be distinguished according to the ambient: in vitro, in fermentation systems, and in soil–plant systems. In fermentation systems and in soil, the process of P-solubilization is well established and attributed to the biochemical activity of microorganisms capable of acidifying the microenvironment as described above. To substitute for the chemical P-fertilizers, a wide number of commercial microbial products are produced and introduced into soils. The well-elaborated scheme for studying and producing P-solubilizing biofertilizers includes various steps starting with isolation, identification and selection of P-solubilizing microorganisms, optimization of the fermentation stage of their production followed by formulation stage, field testing and commercialization [65,66]. However, some recent publications questioned the validity of this concept [67] and particularly the effect of microbial P-solubilization suggesting that soil microorganisms solubilize P just to satisfy their needs. Another key suggestion is that the mechanisms of P-solubilization in vitro and in vivo are different. The main reason for these assumptions is the lack of efficacy of biofertilizers in field conditions. On the other hand, Barrow [68] analyzed the two theories concerning the nature of phosphate in soil considering the adsorption–penetration theory is the only valid theory explaining the fate of phosphate in soil, thus changing the model of P-fertilization to feed the plant not the soil. The theory of these authors should also try to explain the P-cycling in soil with or without microbial metabolites. These suggestions are very important as they will force studies on microbial P-solubilization in soils—a process that seems a bit “frozen”, continuously repeating one and the same research schemes. In fact, the mechanism of the biochemical production of organic acids in microorganisms seems always the same. The concentration of the produced organic acids depends on environmental conditions. However, it is one thing to produce a given metabolite in a fermenter under controlled conditions and a totally different thing to produce the same amount of acid in the soil. Even in controlled conditions, there are a number of parameters that determine the metabolic activity of the microbial producer and the final metabolite yield, such as the amount and age of the inoculum, temperature, pH, aeration, medium composition, etc. [69]. The same is valid in soil conditions where the number of factors that determine the production of whichever plant beneficial metabolite is even higher and for this reason, the positive P-solubilizing effect of the existing or introduced fungal microbial inoculant would be different. Taking all these factors into consideration, it would be easy to understand why some fungal microorganisms behave as over-producers of organic acids in vitro (for example, the industrial concentrations of citric acid produced by *A. niger* can reach 600 mM [70]) although they do not need high organic acid production, or why such fungi solubilize phosphates, additionally supplied to the medium, if they supposedly need P only for their own development. On the other hand, as it is impossible to create the same conditions leading to overproduction of metabolites in soil, the concentration of the latter is low, but in theory sufficient to produce a continuous flow of P-solubilizing agents, produce siderophores, lignocellulose-degrading enzymes, antibiotics, phytohormones, etc. as proved in vitro [71]. The total concentrations of organic anions in the soil solution may appear insufficient to ensure the dissolution of minerals. However, millimolar concentrations could be registered in fungal hyphae microenvironments [72]. Particularly in mycorrhized plants, mycorrhizal hyphae could translocate dissolved minerals directly to the plants bearing also in mind that some mycorrhizal fungi are able to solubilize P in vitro [73].

Another point is that many reports on P-solubilizing microorganisms, especially on P-solubilizing bacteria, attribute P-solubilization to gluconic acid production. Indeed, gluconic acid production results in medium acidification and hence in tricalcium phosphate solubilization in vitro [74]. However, gluconic acid is ineffective to solubilize phosphates since it is a weak acid and has a low capacity to complex cations linked to P [75,76]. Therefore, is very unlikely that gluconic acid would have a major role in P-solubilization in soil due to the pH buffering capacity of soil and its low capacity to promote ligand exchange with soil compounds holding P. Moreover, in acidic soils, the effect of gluconic acid would be null, as demonstrated in solubilization tests with Ca-phosphates under controlled pH [75]. Hence, field trials with P-solubilizing microorganisms in which P-solubilization activity in vitro results from gluconic acid probably will fail, as reported by Meyer et al. [77], who could not find any increase in P-uptake by plants inoculated with a gluconic acid producing *Pseudomonas protegens* and a non-producing mutant as well. Curiously, there is a bias in studies with P-solubilizing microorganisms that put gluconic acid as the main organic acid involved in P-solubilization, including some efforts to obtain overproducing gluconic acid bacteria [78,79,80,81]. On the other hand, oxalic acid—a strong organic acid produced by some P-solubilizing microorganisms, especially fungi, which is as effective as sulfuric acid to release P from rock phosphate [76]—is generally neglected. This acid and the P-solubilizing fungus producing it (*Aspergillus niger*) were capable of desorbing P from a highly weathered soil [64]. Further, the release of oxalic acid is largely accepted as a mechanism by which mycorrhizae increase the rate of soil weathering and enhance the availability of nutrients for plant uptake [82]. Thus, it should be questioned if PSM unsuccess in some field trials is biased by the selection of P-solubilizing microorganisms based on the gluconic acid production capacity.

The whole process of production of plant-available P by soil microorganisms is extremely complex with a wide number of abiotic and biotic factors characterizing each different soil or site. For this reason, there is no universal fungal biofertilizer able to provoke P-solubilization and plant growth enhancement everywhere [83,84,85]. In the latter case, *Penicillium bilaiae* was tested in 27 different soils. As a part of the analysis of prerequisites for selecting an effective biofertilizer in general, we should mention the isolation of local microorganism(s) with specific desired functions, which are reintroduced into the soil would be able to rapidly adapt to the “well-known” existing abiotic and biotic reality; develop an optimized fermentation production system; formulate commercial products containing, at least partly, a sufficient amount of nutrients to facilitate the inoculant establishment in the soil [86] (Figure 2). In this sense, fungal-based biofertilizers, including P-solubilizing ones, demonstrate a number of advantages by being more adaptive to different stress factors [87]. The nature and characteristics of the target plant should also be considered as well as the soil properties, including the microbial profile. If we assume that plants contract microorganisms with specific traits, we should select the same specific plant–microbe combination in field biofertilization. Fungal community structures were shown to be particularly closely related to specific soil characteristics such as available P, exchangeable H^+^ and Al^3+^, NH_4_^+^-N, NO_3_^−^-N, and pH [88]. Therefore, we could not expect the development of the same microbial structure and activity in soil with different specific characteristics. As in crop breeding, target traits (e.g., P-solubilization) and environmental (including host plant) compatibility must be considered side-by-side during biofertilizer development [89]. In summary, the use of plant beneficial microorganisms, including P-solubilizers, could be a process similar to personalized medicine for humans but oriented to “cure” a specific deficiency in a soil–plant system.

## 4. Alternative Approach to Application of P-Solubilizing Fungi

The strong P-solubilizing activity of fungal microorganisms such as *Penicillium*, *Aspergillus*, and *Trichoderma* strains can be used to solubilize P-bearing materials in fermentation systems and substitute for chemical processing. Here, two lines of application could be distinguished: (1) microbial P-solubilization directly in the fermentation systems and (2) P-solubilization by organic acids produced after fermentation processes by fungal microorganisms without the presence of phosphate sources.

Particularly fungal treated (partially solubilized) P-bearing materials can be produced in solid-state fermentation (SSF) processes using agro-industrial wastes as substrates [55,90]. All parameters and culture media could be optimized to facilitate fungal growth, metabolic activity, and P-solubilization [65,90]. The advantage of such a model for P-solubilization is that the resulting final fermentation product contains a partially solubilized P-source, mineralized organic matter (agro-wastes), and fungal mycelium is a formulated biofertilizer that could be applied in the soil directly or after thermal treatment [90]. Similarly, P-solubilization is carried out in submerged fermentation (SmF) where the fungal microorganisms are tested for the production of other metabolites with plant beneficial properties such as siderophores, indole-3-acetic acid (IAA), antibiotics, volatile organic compounds, etc. (Figure 1). The inclusion of additional components in the medium proved a possible increase in organic acid production and P-solubilization efficacy [91]. Applying various types of SmF and different fermenter designs such as fed-batch and repeated-batch fermentations and air-lift bioreactor seems advantageous, increasing both the organic acid production and P-solubilization [92,93,94]. Immobilized cell technology was also applied in SmF with P-solubilizing filamentous fungi combining the advantages of both the immobilization state of the cells and the repeated-batch fermentations [66,95] (Figure 2). In many cases, these biofertilizers demonstrated high P-solubilizing activity but also multifunctional properties, for example, biocontrol [96,97]. Due to the continuous efforts of many scientists, the progress in fungal P-solubilization in fermentation conditions during the last 20–30 years is impressive. Various species of fungal microorganisms were isolated, characterized, selected for their P-solubilizing activity and further experimented within different modes of fermentation processes. The number of studies in conditions of liquid submerged and solid-state fermentation is enormous and impossible to include in this review. In many of these works, fermentation parameters and media were optimized, and metabolite profiles of the respective fungi were determined simultaneously with their P-solubilizing activity.

The obtained results generated the idea of applying the resulting final liquid and free of mycelium products containing solubilized P directly in soil–plant systems as post-biotic fertilizers [7] (Figure 2). Mendes et al. [98] showed the practical advantage of applying such kinds of biofertilizers by analyzing *Trifolium repens* grown in soil or soil-less (vermiculite/perlite) conditions. The application of both filtrated and non-filtrated fermentation liquid samples was found to promote growth and P-uptake of the test plant (*Trifollium repens*), particularly in treatments that received microbially solubilized phosphate. A simple method of P-solubilization and biofertilizer mycelium-free production based on immobilized *Piriformospora indica* employed in a repeated-batch fermentation process was developed in the presence of bone char as a P-source [99]. Thus, the final liquid contained all plant-stimulating metabolites released by the fungus and soluble phosphate. The results demonstrated that by introducing cell-free liquid into the soil–plant system, in addition to the improved plant growth and plant P-content, normally registered in the test plant, other beneficial effects can be observed. Moreover, *A. niger* with proven P-solubilizing activity [100] can promote the growth of vegetable seedlings [101] and can be used as a biocontrol agent against soil-borne plant pathogens *Fusarium oxysporum* f. sp. *phaseoli*, *Fusarium solani* f. sp. *phaseoli*, *Macrophomina phaseolina*, *Rhizoctonia solani*, *Sclerotinia sclerotiorum*, *Sclerotium cepivorum*, and *Sclerotium rolfsii* (unpublished). This effect was attributed to the release of thermostable metabolites and volatile organic compounds (VOC). The production of VOC is of particular interest as an important part of the overall effect of these metabolites released by biofertilizers and plants, which determine the sensory perception of fruits [102]. Therefore, it is a good point to mention that when fungal biofertilizer actions are assessed, a wide number of beneficial side effects should be evaluated (Figure 1) including the quality of the target plant and the production of high-quality functional foods [103]. At the European level there is a serious interest in the development of tools able to predict the functionality of agri-foods and their organoleptic characteristics, starting from strategies based on biofertilizers and thus avoiding chemical fertilizers, through enhancement of antioxidants, vitamins, minerals, flavors, and other human health-related compounds which could make our diet healthier and free from chemicals (see the web page of the EC Project VIRTUOUS—http://virtuoush2020.com, accessed on 20 July 2022). In this line of research, another possibility for P-solubilization should be mentioned such as the solubilization of P-bearing materials out of the fermenters by organic acids produced by microbial P-solubilizers. The high potential of oxalic acid for rock phosphate solubilization was frequently shown [57,104]. Recently, Mendes et al. [59,76] showed that this di-carboxylic acid is able to extract 100% of P contained in rock phosphates with different characteristics. Moreover, oxalic acid is more efficient than sulfuric acid, releasing more P per mol of acid applied [76]. Thus, mycogenic oxalic acid appears to be a promising alternative for the production of phosphate fertilizers thus substituting the traditionally used sulfuric acid offering an efficient, low-cost, and environmentally friendly method for P-extraction [76,105].

## 5. Conclusions

In the few last years, the search for substitutes for phosphate fertilizers is of high importance and urgency as the natural P-source (rock phosphate) is a finite resource. Other reasons also include public acceptance of bio-based agricultural products, climate change, and the growing population. In controlled conditions (laboratories and greenhouses), a large number of microorganisms were reported to solubilize inorganic and organic phosphates. However, the biotech companies were not focused on mass-producing bio-based fertilizers due to their high cost when compared with the low prices of mineral fertilizers. Particularly at this moment, the situation is different: due to the crisis provoked by the war in Ukraine, the prices of chemical fertilizers are extremely high. On the other hand, the success of the field application of biofertilizers is not always visible. The reasons for this are mainly the wrong schemes of product development and a lack of collaboration between experts in different fields of research. Wrong experiments in soil without any microbial inoculant development/formulation could be the reason for a “new” theory or call for reconsideration of the current status. The implementation of new technologies and high investments are not sufficient for the development of serious biotechnological production of P-biofertilizers. What is needed is the establishment of expert protocols for each step of the P-biofertilizer production, which should include all potential possibilities and risks (Figure 2). Alternative P-sources should be tested including low-P rocks. In soil and out of soil microbial solubilization should be considered depending on previous deep analysis of variants. Multidisciplinary approaches and tools including soil science, microbiological, biotechnological, and plant physiology analysis, artificial intelligence, machine learning, and mist computing should be widely used to predict, select and control plant (P) nutrition, particularly in field conditions based on biofertilizers (and precision agriculture in general) and create a larger view on the effects of the multiple properties of the fungal microorganisms from seed germination to fruit quality. All this is following a strategy similar to personalized medicine for humans but is oriented to “cure” a specific deficiency in a soil–plant system. Multiple abiotic and biotic soil factors and plant characteristics and plant–soil–microbe history should be considered before taking a final management decision. This diversity of variables means that one approach for the production and application of fungal P-solubilizers cannot fit all the different contexts, but that different and interconnected strategies should be investigated (Figure 2). Based on the scientific literature, a strong wave of novel strategies and wider application of fungal biofertilizers are expected in the near future including in the field of fungal P-solubilization, always looking for safe and healthy final products [106,107].

## Figures and Tables

**Figure 1 microorganisms-10-01716-f001:**
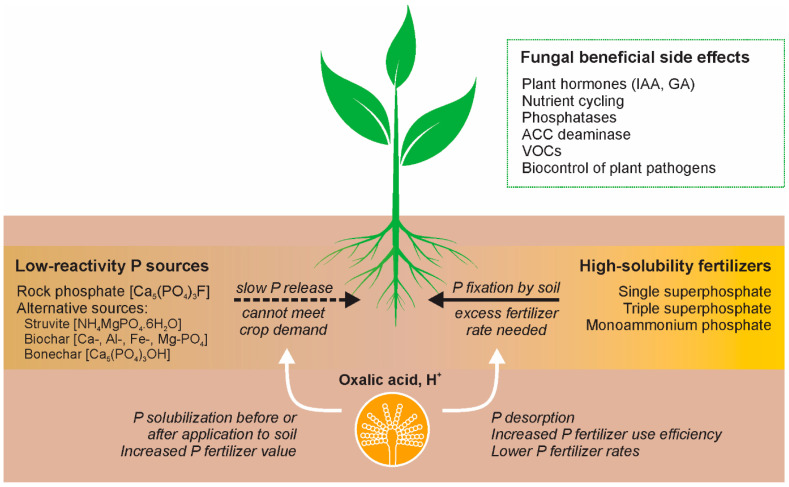
**Fungal P-solubilization and potential beneficial side effects on plants.** Fungal acidification by H^+^ extrusion and organic acid production can improve plant access to P by the solubilization of low-reactivity P-sources and P-desorption from soil minerals. In the first case, phosphate-solubilizing fungi (PSF) can increase the fertilizer value of low-reactivity P-sources by treatment before application to the soil or by direct inoculation to the soil fertilized with the source. In the second scenario, PSF can revert P-fixation by soil, improving the fertilizer use efficiency and allowing reducing P-fertilizer rates. Moreover, PSF inoculation can promote plant growth by additional mechanisms, such as nutrient cycling, biocontrol of plant pathogens, synthesis of plant hormones such as indoleacetic acid (IAA) and gibberellins (GA), volatile organic compounds (VOCs), and enzymes.

**Figure 2 microorganisms-10-01716-f002:**
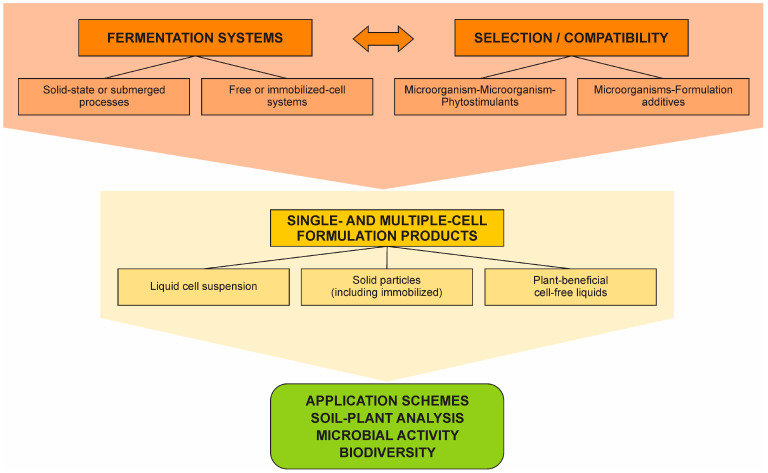
**Development of microbial products for plant nutrition.** Before reaching field tests, a potential beneficial microorganism must be characterized, efficiently produced, and correctly formulated. The choice of fermentation systems, mixtures of microorganisms and/or phytostimulants and additives, and the formulation will depend on factors such as the microbial species, target plant, mode of action, mode of application, etc. Any wrong decision can compromise the microbial product efficiency.

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
