# Peer review of "Fungi, P-Solubilization, and Plant Nutrition"

_microorganisms, 2022, doi:10.3390/microorganisms10091716_

Round 1

Reviewer 1 Report

This review article discusses the efficient use of environmental phosphorus resources by plants from the perspective of microbial(fungal) functions. Because of the wide range of topics covered, in my opinion, it is not clear what kind of readers this paper targets and what areas of research it attempts to summarize.

Minor concerns

lines 59-61, It is hard to imagine a situation where microorganisms help plants absorb nutrients where soil organic matter is low. What is the difference between soil microbes and soil organic matter? Is it possible for soil microorganisms to work ( proliferate) where there is little soil organic matter? In the rhizosphere? Throughout the text, there is a critical lack of biological review of microbial functions, including the uneven distribution of organic matter in the soil, interactions with plants, nutrient availability, and the physiological ecology of microorganisms.

line 74, Please cite references that indicate that bacteria grow in the mycelosphere.

lines 88-97, This sentence is also used in the summary. I think it should be removed.

lines 111-139, These sentences explain common knowledge and a major cause of blurring the focus of the discussion as a review article related to microorganisms.

Author Response

This review article discusses the efficient use of environmental phosphorus resources by plants from the perspective of microbial(fungal) functions. Because of the wide range of topics covered, in my opinion, it is not clear what kind of readers this paper targets and what areas of research it attempts to summarize.

Authors: We thank the Reviewer for the comments. We improved the manuscript following the suggestions.

Minor concerns

lines 59-61, It is hard to imagine a situation where microorganisms help plants absorb nutrients where soil organic matter is low. What is the difference between soil microbes and soil organic matter? Is it possible for soil microorganisms to work ( proliferate) where there is little soil organic matter? In the rhizosphere? Throughout the text, there is a critical lack of biological review of microbial functions, including the uneven distribution of organic matter in the soil, interactions with plants, nutrient availability, and the physiological ecology of microorganisms.

Authors: The reviewer is right. The following lines were deleted to avoid further discussion: lines 59-61….”In soils poor in organic matter and nutrients such as nitrogen and phosphorus, these soil microorganisms are considered as regulators of plant productivity and are shown to be responsible for acquisition of these macro-elements by plants.”

As for the “critical lack of biological review of microbial functions”, you are right. However, we consciously tried to avoid such kind of deep discussion – our intention was to pay attention on fungi as P-solubilizers, the mechanism of P-solubilization, alternative P-sources, alternative P production through fermentation, production and formulation of fungal P biofertilizers.

line 74, Please cite references that indicate that bacteria grow in the mycelosphere.

Authors: Ref 14 Zhang Y, Kastman EK, Guasto JS, Wolfe BE (2018) Fungal network shape dynamics of bacterial dispersal and community assembly in cheese rind microbiomes. Nature Communications. DOI: 10.1038/s41467-017-02522-z

was eliminated and a new one, more related to the text, was included but substitute Ref 13:

  1. Jiao Sh, Chu H, Zhang B, Wei X, Chen W, Wei G (2022) Linking soil fungi to bacterial community assembly in arid ecosystems. iMeta, 1:e2. https://doi.org/10.1002/imt2.2

lines 88-97, This sentence is also used in the summary. I think it should be removed.

Authors: Yes, but we kindly ask to use this mode of describing the aim of the work in both the abstract and in the purpose of the review at the end of the Introduction.

lines 111-139, These sentences explain common knowledge and a major cause of blurring the focus of the discussion as a review article related to microorganisms.

Authors: Yes, you are right. As you are expert in the field, it sounds primitive, but we kindly ask for your approval of this part as we include all types of P, conventional (RP) and alternative, in our analysis of their possible use after fermentation processes. On the other hand, it would be necessary to use this trivial information just to explain the need of alternative P sources. We removed some information for conciseness. Please see the updated text.

Reviewer 2 Report

Authors provide a nice review of the P and fungi. However, it needs a thorough revision on the grammar, style, and formatting. 

Please include https://doi.org/10.1016/j.gloenvcha.2008.10.009 in the introduction.

Need to add more information on the impact of pyrolysis temperature on P in the biochar/bonechar section. Not all biochar are same, please include the information in the following paper: 

https://doi.org/10.1016/j.geoderma.2016.04.020

Mycorrhizal fungi is one of the fastest growing biofertilizer business worldwide. Authors need to elaborate on mycorrhizal fungi and P. 

Authors have included role of bacteria in P solubilization, manuscript could be improved by including more on phosphorus solubilizing bacteria, the synergy between fungi and bacteria. And maybe change the title to include both?

Please see:

https://doi.org/10.1038/s41598-019-40910-1

Author Response

Authors provide a nice review of the P and fungi. However, it needs a thorough revision on the grammar, style, and formatting. 

Authors: We have performed a careful review of the grammar, style, and formatting.

Please include https://doi.org/10.1016/j.gloenvcha.2008.10.009 in the introduction.

Authors: Suggested article was included as Ref 23

Need to add more information on the impact of pyrolysis temperature on P in the biochar/bonechar section. Not all biochar are same, please include the information in the following paper: 

https://doi.org/10.1016/j.geoderma.2016.04.020

Authors: Included as Ref 52: “It should be noted, that when assessing the role of biochars as potential P-sources and soil P improvers, we should also distinguish between different types of these products formed at different temperatures, 500-700o C being the most appropriate (49,52).” See lines 211-213. 

Mycorrhizal fungi is one of the fastest growing biofertilizer business worldwide. Authors need to elaborate on mycorrhizal fungi and P. 

Authors: Yes, but Mycorrhizal fungi are a whole world. On the other hand, given the recent critical reconsideration of their role in P nutrition and the fact that a group of scientist consider Mycorrhizae as a root extention without direct effect on P solubilization, we decided to not include this important group in our work. Moreover, a great part of the paper concerns fermentations.

Authors have included role of bacteria in P solubilization, manuscript could be improved by including more on phosphorus solubilizing bacteria, the synergy between fungi and bacteria. And maybe change the title to include both?

Authors: In this review we really like to call more attention to fungal P solubilization, once this important group has received less attention as a plant-beneficial microorganism.

Round 2

Reviewer 1 Report

The authors have responded to all the review comments raised.

Reviewer 2 Report

Thank you for revising the manuscript. There are several grammatical errors in the manuscript. Please do a thorough review.